# Danazol as a Treatment for Uterine Arteriovenous Malformation: A Case Report

**DOI:** 10.3390/jpm13091289

**Published:** 2023-08-23

**Authors:** Hyunjin Tak, Kyong-No Lee, Ji-Won Ryu, Keun-Young Lee, Ga-Hyun Son

**Affiliations:** 1Department of Obstetrics and Gynecology, Hallym University College of Medicine, Kangnam Sacred Heart Hospital, Seoul 07441, Republic of Korea; hjtak23@gmail.com (H.T.); koala1115@naver.com (J.-W.R.); mfmlee@hallym.ac.kr (K.-Y.L.); 2Department of Obstetrics and Gynecology, Chungnam National University Hospital, Daejeon 35015, Republic of Korea; kyongnolee@cnuh.co.kr; 3Institute of New Frontier Research Team, College of Medicine, Hallym University, Chuncheon 24252, Republic of Korea

**Keywords:** arteriovenous malformation, embolization, danazol, systemic review

## Abstract

Uterine arteriovenous malformation (AVM) is associated with a risk of massive uterine bleeding. Although uterine artery embolization remains the first-line treatment for AVM, there has been a recent exploration of pharmacological options. Danazol is known to reduce blood flow to the uterus; however, our understanding of its therapeutic efficacy for AVM remains limited. Herein, we present the results of danazol use in patients with uterine AVM. We retrospectively reviewed the medical records of patients who received danazol for the treatment of AVM between January 2013 and November 2022. The cohort comprised 10 patients who developed AVM after dilatation and curettage (D&C), abortion, or cesarean section. Danazol was administered twice daily at a total dose of 400 mg/day, and was employed for AVM treatment in hemodynamically stable patients who provided consent and were devoid of massive bleeding. Outpatient follow-ups (ultrasound measurements of AVM size and symptom assessment) were performed every 2 weeks. AVM was successfully treated with danazol in most patients with no adverse event. Eight postabortal patients had complete resolution of AVM after an average of 45 days (range 14–70 days). Of two patients who developed AVM after a cesarean section, one experienced AVM reduction, and the other developed massive bleeding, requiring emergency uterine artery embolization. In light of these outcomes, danazol can be potentially prioritized over uterine artery embolization in the treatment of AVM after abortion in hemodynamically stable patients.

## 1. Introduction

Uterine arteriovenous malformation (AVM) is a vascular structural anomaly caused by abnormal connections between the uterine arteries and veins that bypass the capillary system in the uterine myometrium. Despite its rarity, AVM can be life-threatening due to the risk of massive uterine bleeding [1,2,3]. While some cases of uterine AVM are congenital, most develop after injuries to the uterine tissues, such as those occurring during dilation and curettage (D&C), abortion, vaginal delivery, and cesarean section [4,5,6,7].

Symptoms of AVM include menorrhagia, abnormal/irregular bleeding, anemia, and lower abdominal pain. As massive bleeding can induce hemodynamic instability, emergency hysterectomy may be considered in severe cases [2,8]. However, since AVM predominantly occurs in women of reproductive age, fertility preservation is an important consideration when determining treatment.

In the past, AVM was pathologically diagnosed following hysterectomy in most cases. Currently, various techniques such as contrast-enhanced computed tomography, hysteroscopy, angiography, and magnetic resonance imaging are used for the diagnosis of AVM. Recently, vaginal sonography combined with color Doppler imaging has emerged as the preferred initial choice of diagnostic method due to its benefits of being rapid, simple, cost-effective, and noninvasive. This technique allows visualization of grayscale cystic lesions and tubular anechoic spaces within the myometrium, coupled with the identification of turbulent hypervascular flow via Doppler imaging [9,10].

There is a scarcity of data concerning the optimal management of AVM; historically, uterine artery embolization (UAE) is commonly regarded as a first-line treatment [11,12]. A key advantage of UAE is that it effectively resolves AVM while preserving the uterus; however, it also has several drawbacks, such as being costly, invasive, and not universally available since it requires hospitalization. In addition, there are concerns that UAE can adversely impact the patient’s fertility, given its associations with ovarian insufficiency and uterine synechiae [13]. As such, various pharmacological approaches have been attempted in recent years, including hormonal suppression with gonadotropin-releasing hormone (GnRH) agonists, uterotonic agents such as methylergonovine maleate, as well as chemotherapeutic methods (methotrexate), and synthetic androgen such as danazol [14,15,16,17,18,19,20]. These treatments require prolonged administration, thus contributing to an increased risk of massive bleeding during the treatment period, but their growing prescription is due to their enhanced accessibility, affordability, and reduced invasiveness. Furthermore, they have the advantage of not affecting future fertility. The persistent interest in pharmacological approaches has led to a continuous series of studies focused on the management of AVM cases using pharmacological methods, with most of these methods yielding favorable outcomes [14,15,16,17,18,19,20].

Danazol, an isoxazole derivative of 17-alpha-ethinyl testosterone, has previously been used to treat dysfunctional uterine bleeding or heavy menstrual bleeding. Although the exact mechanism of action is not known, it is thought to be effective in reducing uterine artery blood flow [21]. To date, two cases of the use of danazol for AVM treatment with successful results have previously been reported. However, due to a lack of data, the efficacy of danazol remains insufficient [16]. In the present study, we present the results of danazol use for AVM treatment in 10 patients.

## 2. Case Series

In this retrospective study conducted from January 2013 to November 2022, we enrolled 10 patients to receive danazol for the treatment of AVM at Kangnam Sacred Heart Hospital. This study received approval from the Institutional Review Board (approval number: 2023-02-003), which waived the requirement for informed consent. 

AVM was diagnosed by transvaginal sonography with color Doppler imaging based on the presence of hypoechoic or tubular anechoic areas within the myometrium, with abundant high flow rates and low resistance on color Doppler imaging. To eliminate the possibility of vasculogenesis associated with gestational trophoblastic disease, the serum human chorionic gonadotrophin (hCG) levels of patients were evaluated before treatment. Patients with serum hCG levels of ≥30 mUI/mL were monitored to confirm a decrease in these levels.

Among the available pharmacological choices, danazol was selected due to its established use in managing menorrhagia and its effectiveness in decreasing uterine blood flow [21]. Moreover, danazol was preferred due to its affordability, cost-effectiveness, and ease of use.

Danazol administration was considered as a treatment option for AVMs with a maximum diameter of <5 cm in hemodynamically stable patients without massive bleeding, whereas UAE was preferably performed in patients with massive bleeding or those with hemodynamic instability. In addition, factors such as future pregnancy plans and patient compliance were taken into consideration, as consistent medication and regular hospital visits are crucial aspects of pharmacological treatment. Adequate information was provided to the patients on the possibility of treatment failure, including the possibility of emergency UAE or hysterectomy owing to significant bleeding during treatment or treatment failure, and the use of other available medical treatments other than danazol. Patients were thoroughly informed about the potential adverse effects of danazol such as weight gain, acne, voice change caused by the androgen effect, and menopause-like symptoms including hot flashes and vaginal dryness [22]. Danazol was prescribed to patients who consented to its administration, with a recommended dosage of 400 mg/day divided into two doses, taken twice daily. 

In total, danazol therapy was initiated in 16 patients who received sufficient explanation. Of these 16 patients, 6 were lost during the follow-up period. Thus, we analyzed the treatment progress of the remaining 10 patients who successfully completed the prescribed treatment. These patients were instructed to undergo regular outpatient follow-ups every 2 weeks for ultrasound measurements of AVM size, assessments of symptoms such as vaginal bleeding, and to check signs of adverse effects of danazol. The decision to continue with danazol therapy was based on these findings. Danazol therapy was discontinued once symptoms of vaginal bleeding were resolved and ultrasound examinations confirmed the absence of visible flow and revealed normal appearance in both the endometrium and myometrium.

Seven of the ten analyzed patients had developed AVM after D&C for abortion, one patient had developed AVM after complete abortion, and two had developed AVM following cesarean section. Detailed clinical characteristics of these patients are presented in Table 1. All patients presented to the hospital after developing vaginal bleeding within 6 weeks. None of the patients were under treatment for hypertension or coagulation disorders. At the time of their hospital visits, transvaginal ultrasound was performed for all patients, leading to the detection of AVM. The average AVM size upon detection was approximately 3.2 × 1.6 cm. The location of the AVM varied across patients: of ten AVMs, six were located in the posterior wall of the uterus, whereas four were located in the anterior wall of the uterus. Danazol was administered in an average of 44.5 days (range, 24.0–67.0 days) after the event, and the mean duration of danazol therapy was 42.0 days (range, 14.0–70.0 days) on average.

Ultrasound findings revealed complete resolution of AVM following danazol treatment in eight patients, all of whom developed AVM after abortion (either D&C or complete abortion). All eight of these patients experienced a return to normal menstruation within 2 months after completion of danazol treatment (Figure 1). 

In one patient who underwent cesarean section due to monochorionic twin pregnancy at 36 weeks, although blood flow in the AVM disappeared on ultrasound after using danazol for 35 days, a 3.1 × 1.0 cm hyperechogenic lesion was observed around the endometrium. Danazol administration was discontinued since the absence of any AVM symptoms such as vaginal bleeding. During the 6-month follow-up, the size of the hyperechogenic lesion decreased to <1 cm and the patient had normal menstruation without any abnormal symptoms, resulting in follow-up loss during the observation period without further treatment. Another patient had undergone a cesarean section at 37 weeks due to placenta previa totalis. Accompanied by placenta accreta, the placenta was delivered with difficulty during surgery. This patient was diagnosed with AVM on the 24th day after surgery, and during danazol treatment, significant vaginal bleeding continued; thus, UAE was performed on day 21 of danazol treatment. 

None of the other patients developed massive hemorrhage during danazol treatment, and the volume of bleeding tended to drop over the duration of danazol therapy, eventually ceasing altogether. In addition, no patients developed adverse effects related to danazol.

## 3. Discussion

Although there are several therapeutic options for uterine AVM, UAE is generally considered the first-line treatment. However, UAE may not be the optimal treatment for women of childbearing age, as it is costly, invasive, may not be universally available, and carries concerns regarding future fertility associated with post-procedural intrauterine adhesions or ovarian insufficiency. In contrast, pharmacological treatments have the advantages of being less expensive, easier to access, less invasive, and not likely to affect future fertility. However, pharmacological treatment can only be applied to patients who are hemodynamically stable and do not have a massive hemorrhage. As many cases of AVM are detected early owing to advances in and increased access to ultrasound, most recent patients are hemodynamically stable without massive bleeding at the time of diagnosis. As such, the number of patients to which the pharmacological approach can apply is increasing.

Pharmacological treatments that have been reported for AVM include GnRH agonists, progestins, danazol, uterotonics, methotrexate, and combinations of these drugs [14,15,16,17,18,19,20]. A recent meta-analysis review of AVM treatment reported that the overall success rate of pharmacological treatment for AVM was 88% and that the most studied pharmacological therapies were GnRH agonist and progestin, both of which showed high efficacy with low complication rates [14].

Vilos et al. showed that the combination of a GnRH agonist with a transient aromatase inhibitor and tranexamic acid was effective in both treating and preserving reproductive capabilities in women with AVMs accompanied by abnormal uterine bleeding [15]. Despite the limited amount of data available on the use of GnRH agonists for AVM treatment, GnRH agonists can be effective in managing abnormal uterine bleeding linked to AVMs due to the significant hypoestrogenic state induced by GnRH agonists. Nevertheless, the duration of using the GnRH agonist was relatively long, ranging from 3 to 11 months [15]. Taneja et al. suggested that oral norethisterone, a derivative of progesterone, induced stromal decidualization and subsequent thinning of the endometrium in the postabortal state. This mechanism is believed to prevent the shedding of the endometrium, effectively averting exposure to AVM. In a study involving 30 patients, despite the first 3-week course of therapy, abnormal bleeding continued in 13 patients, necessitating a second course [17]. Consequently, further research is necessary to evaluate the safety and efficacy of progestin therapy. The potential of danazol as a treatment method for AVM with promising outcomes was demonstrated in two other case reports prior to this study; however, the limited number of participants in these studies hinders a comprehensive evaluation of its effectiveness [16,21]. Therefore, in this study, danazol was selected with the expectation that it could stably treat AVM in a short period of time.

Also, as these success rates mentioned above are mainly based on case reports or case series of successful outcomes, this success rate may not be accurate, and comparisons cannot be made with the success rate of UAE. In addition, as AVM is rare and there is consequently a lack of data, it is difficult to identify the factors related to treatment success because of the significant heterogeneity of patient characteristics and pharmacological treatment regimen and doses.

This study reported the experience of using one drug in a relatively large number of patients with AVM treated in a single center. In the present study, AVMs completely resolved in 8 of 10 patients prescribed danazol. In one patient, the AVM did not completely resolve but significantly decreased in size, and the residual AVM did not trigger any events during the follow-up period of seven months after medication administration. One other patient developed a significant hemorrhage after 21 days of danazol treatment, necessitating UAE. In both cases of incomplete resolution, patients developed AVMs after undergoing a cesarean section, indicating that danazol could be less effective in patients who have had a cesarean section. The strength of this study lies in its relatively large number of patients with AVM and its finding that danazol was effective in postabortal patients. However, this study has some limitations. First, it lacks a control group, such as one using alternative pharmacological treatments, or an expectant management. Thus, the superior efficacy of danazol for AVM treatment could not be evaluated through group comparisons. Second, the diagnosis of AVM was solely based on the presence of a distinct vascular network within the myometrium observed on vaginal sonography combined with color Doppler without using other diagnostic methods such as computed tomography, hysteroscopy, or magnetic resonance imaging, which may be insufficient for diagnosing AVM. Given these limitations, it is essential to incorporate a well-designed control group utilizing a wider range of diagnostic methods in future studies to reveal the effectiveness of danazol in the treatment of AVM.

## 4. Conclusions

The study findings suggest that for the treatment of AVM occurring after D&C or abortion in hemodynamically stable patients without significant massive bleeding, danazol can be preferably administered before considering UAE. Among the ten patients included in this study, eight showed complete resolution of AVM. However, patients must be adequately informed of the possibility of treatment failure and risks of massive bleeding during danazol use before initiating medication. In addition, danazol treatment must be attempted only in patients who display good treatment adherence and are capable of undergoing careful follow-up. Although the results of this study cannot conclude the treatment efficacy of danazol on AVM treatment, they contribute to the growing evidence in the literature of its efficacy and help clinicians in deciding treatment options for AVM.

## Figures and Tables

**Figure 1 jpm-13-01289-f001:**
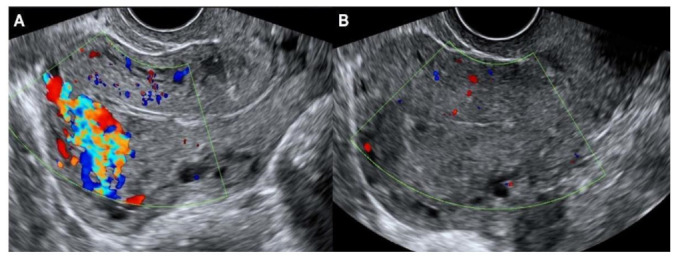
Ultrasound images (**A**) before and (**B**) after danazol therapy in a patient who developed AVM after dilatation and curettage for abortion.

**Table 1 jpm-13-01289-t001:** AVM characteristics and follow-up data.

Patient	Event	Age	Parity	Abortion	Event to AVM Diagnosis (Days)	Danazol Use Period (Days)	AVM Size at Diagnosis (W × L, cm)	AVM Size after Treatment (W × L, cm)	Danazol Start to Last Follow-Up (Days)
1	Abortion, D&C	34	2	1	34	42	4.0 × 1.4	-	28
2	Complete abortion	36	0	2	25	70	2.1 × 1.6	-	52
3	Cesarean section	38	0	2	59	35	4.4 × 2.5	3.1 × 1.0	42
4	Abortion, D&C	35	0	0	26	14	2.3 × 1.7	-	19
5	Abortion, D&C	31	0	1	62	49	2.9 × 1.6	-	77
6	Abortion, D&C	37	1	1	51	42	3.6 × 2.6	-	30
7	Abortion, D&C	34	0	1	38	70	2.7 × 1.6	-	77
8	Abortion, D&C	37	1	3	67	70	4.2 × 1.9	-	83
9	Cesarean section	31	1	1	24	21	4.8 × 2.0	0.8 × 0.5	136
10	Abortion, D&C	41	2	3	58	42	1.2 × 0.9	-	365

AVM: arteriovenous malformation; D&C: dilatation and curettage; W: width; L: length.

## Data Availability

The data presented in this study are available on request from the corresponding author. The data are not publicly available due to patients’ privacy.

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
