# Peer review of "Danazol as a Treatment for Uterine Arteriovenous Malformation: A Case Report"

_jpm, 2023, doi:10.3390/jpm13091289_

Round 1

Reviewer 1 Report

The conceptualization of the manuscript, evaluating the results of a medical therapy by Danazole in the management of AVMs, is valuable.

Nevertheless, the study is retrospective and no control harm is provided. The diagnostic procedure (lines 67-72) such as described is insufficient to support a diagnosis of AVM. Neither TAC, MRI or Angiography were considered in the diagnostic work-up. The level of βHCG is of no value in excluding RPOC or adherent placenta.

The Table 1 must be re-written.

The English grammar and style must be substantially reviewed by a native English speaker confident with medical reports.

The English grammar and style must be extensively reviewed

Author Response

RESPONSES TO REVIEWER COMMENTS

# Reviewer 1

Nevertheless, the study is retrospective and no control harm is provided. The diagnostic procedure (lines 67-72) such as described is insufficient to support a diagnosis of AVM. Neither TAC, MRI or Angiography were considered in the diagnostic work-up. The level of βHCG is of no value in excluding RPOC or adherent placenta.

Reply: Thank you for bringing this to our attention. We have included information about the diagnostic method for AVM (line 44-51). Sonographic findings are often nonspecific and may not be sufficient for an accurate AVM diagnosis. To enhance specificity, we employed color Doppler ultrasound, which offers a more distinct image by depicting a color mosaic pattern with thickened vessels and low reversals. Whereas transvaginal sonography with color Doppler imaging may have its limitations, ultrasound was chosen as a diagnostic method considering factors such as cost-effectiveness and ease of examination. We have eliminated the sentence concerning β-hCG in relation to RPOC and adherent placenta.

The Table 1 must be re-written.

Reply: Thank you for your suggestion. We have re-edited the table to make it easier to read by subtracting β-hCG related information..

The English grammar and style must be substantially reviewed by a native English speaker confident with medical reports.

Reply: Thank you for addressing this issue. We have extensively edited English grammar and styles throughout the manuscript by native English speaker.

Reviewer 2 Report

In the review "Danazol as a treatment for uterine arteriovenous malformation:  A case series"  by Hyunjin Tak et al.:

It is a good paper 

I suggest adding more introduction, this implies adding current bibliographic references, most of them are very old.

Likewise with the discussion, it is very brief, the results give more information.

Highlight very punctually, the conclusions.

 Minor editing of English language required

Author Response

RESPONSES TO REVIEWER COMMENTS

# Reviewer 2

I suggest adding more introduction, this implies adding current bibliographic references, most of them are very old.

Reply: We agree with the reviewer’s advice. We have included updated references on pharmacological treatment for AVM (reference 9,10,12,20)

Likewise with the discussion, it is very brief, the results give more information.

Reply: Thank you for your suggestion. As suggested we have added more findings in results (line 123-125), and also included more information about treatment of AVM in the discussion section (line 169-161).

Highlight very punctually, the conclusions.

Reply: We have revised the first sentence of conclusion into sentence highlighting the results. (line 208-211)

Reviewer 3 Report

The authors present a case series of uterine MAV treated conservatively through danazol administration. 

This article is a welcome addition to the sparse literature already in existence on the subject.  Though the lot size is small it is both a rare pathology and the treatment indications tend to veer to more aggressive practices.  With new information coming to light about potential adverse effects of UAE, this is a welcome alternative, although insufficiently studied.

1. Is this a retrospective study; it is not clearly mentioned in the article. 

2. If not, what other parameters (size, age, parity, etc) were used to select patients for this treatment versus UAE, besides massive bleeding and hemodynamic instability?

3. Were flow rates in uterine arteries measured - confirming danazol effect on blood flow to the uterus?

I would also be curious as to why danazol was preferred over other pharmacological means within your institution.

As a side note, while research for this article is extensive, and the literature is quite sparse, there are also newer publications that can prove the point as well. 

The article is well written and well presented.

Author Response

RESPONSES TO REVIEWER COMMENTS

# Reviewer 3

  1. Is this a retrospective study; it is not clearly mentioned in the article. 

Reply: We thank the reviewer for raising this point. We have accordingly revised the sentence “This study enrolled patients who received danazol for the treatment of AVM between January 2013 and November 2022 at the Kangnam Sacred Heart Hospital.” to “In this retrospective study conducted between January 2013 and November 2022, 10 patients were enrolled to receive danazol for the treatment of AVM at Kangnam Sacred Heart Hospital.”

  1. If not, what other parameters (size, age, parity, etc) were used to select patients for this treatment versus UAE, besides massive bleeding and hemodynamic instability?

Reply: Thank you for addressing this issue. We have taken into account your input and have included the information regarding the decision-making process for the use of pharmacological treatment, primarily focusing on the presence of massive bleeding and hemodynamic instability. Additionally, we clarified that UAE was typically employed for AVM sizes larger than 5 cm. The choice of treatment method was also influenced by factors such as the patient's future pregnancy plans and their compliance with the treatment regimen. These details have been incorporated into the method section (line 92-97).

  1. Were flow rates in uterine arteries measured - confirming danazol effect on blood flow to the uterus?

Reply: We thank the reviewer for their remarks on our study. Since this study is a retrospective study and has been conducted nearly10 years, peak systolic velocity was not measured in all patients, which is the weakness of this study. In this study, the treatment endpoint was defined as the absence of visible flow and the presence of a normal appearance in both the endometrium and myometrium as determined by ultrasonography (line 113-115). Thus, the effect of danazol was assessed by AVM size rather than blood flow velocity.

  1. I would also be curious as to why danazol was preferred over other pharmacological means within your institution.

Reply: Thank you for pointing this out. We selected danazol as the pharmacological method for AVM treatment among various options due to its established use in reducing vaginal bleeding and its effectiveness in decreasing uterine blood flow. In addition, danazol is characterized by its affordability, cost-effectiveness, and user-friendly administration. We have provided the mentioned reasons for opting for danazol in the method section (line 95-97).

  1. As a side note, while research for this article is extensive, and the literature is quite sparse, there are also newer publications that can prove the point as well. 

Reply:  We agree with you and thank you for your suggestion. We have incorporated the updated references pertaining to pharmacological treatment for AVM.(reference 9,10,12,20)

Round 2

Reviewer 1 Report

Dear Author,

The manuscript lack a control harm (ie simply an expectant management)

A diagnosis of AVM based only on Transvaginal Ultrasound may be misleading

I believe these concepts must be underlined in Discussion section 

Author Response

RESPONSES TO REVIEWER COMMENTS (2)
# Reviewer 1
The manuscript lack a control harm (ie simply an expectant management)
A diagnosis of AVM based only on Transvaginal Ultrasound may be misleading
Reply: Thank you for your helpful comment. Since this study is a retrospective study, it has limitations as you mentioned. Therefore, we have added these contents to the discussion section.  (line 207-218)